# Effect of Different Corticosteroid Regimens on the Outcome of Severe COVID-19-Related Acute Respiratory Failure. A Retrospective Analysis

**DOI:** 10.3390/jcm10214847

**Published:** 2021-10-21

**Authors:** Michele Umbrello, Paolo Formenti, Stefano Nespoli, Eleonora Pisano, Cecilia Bonino, Stefano Muttini

**Affiliations:** 1SC Anestesia e Rianimazione II, Ospedale San Carlo Borromeo, ASST Santi Paolo e Carlo, 20153 Milan, Italy; stefano.nespoli.perri@gmail.com (S.N.); stefano.muttini@asst-santipaolocarolo.it (S.M.); 2SC Anestesia e Rianimazione, Ospedale San Paolo–Polo Universitario, ASST Santi Paolo e Carlo, 20153 Milan, Italy; 3Scuola di Specializzazione in Medicina D’emergenza-Urgenza, Università degli Studi di Milan, 20153 Milan, Italy; eleonora.pisano@icloud.com (E.P.); c.boninogru@gmail.com (C.B.)

**Keywords:** acute respiratory distress syndrome (ARDS), COVID-19, critically ill patients, corticosteroids

## Abstract

Background: Systemic corticosteroids are associated with reduced mortality in COVID-19-related acute respiratory failure; however, the type and dose has not yet been established. Objectives: To compare the outcomes of dexamethasone vs. methylprednisolone, along with the effects of rescue, short-term, high-dose boluses of corticosteroids. Methods: Before/after and case/control retrospective analysis of consecutive critically ill COVID-19 subjects. Subjects were initially given dexamethasone; however, after review of the local protocol, methylprednisolone was suggested. A three-day course of 1000 mg/day of methylprednisolone was administered in the case of refractory hypoxemia within the first 10 days of treatment. Propensity score-adjusted comparisons were performed. Results: A total of 81 consecutive subjects were included (85% males, 60 ± 10 years, SAPS II 27 ± 7, SOFA 4 [IQR 3, 6] points) and 51 of these subjects (62.9%) received dexamethasone and 29 (35.8%) had methylprednisolone. The groups were well matched for age, comorbidities, and severity at admission. No differences were found in the duration of ICU stay, hospital mortality, or infectious complications between the groups. A total of 22 subjects (27.2%) received a rescue bolus; these subjects had a significantly lower oxygenation, a higher driving pressure, and an increased ventilatory ratio during the first ten days. Short-term/high-dose boluses were associated with higher hospital mortality, longer mechanical ventilation and ICU and hospital stay, and more infectious complications. A subgroup of subjects who received the boluses had significantly improved oxygenation and lower hospital mortality. Conclusions: We were unable to find any difference between dexamethasone or methylprednisolone on the explored outcomes; high-dose boluses of corticosteroids were associated with a worse outcome. However, a subgroup of subjects was identified in whom the high-dose boluses seemed beneficial.

## Highlights

The exact type and dose of corticosteroid to be used in critically ill patients with COVID-19-related acute hypoxemic respiratory failure has not yet been established.In a retrospective trial, we were unable to find any difference between dexamethasone and methylprednisolone at standard doses in patient-centered outcomes. High-dose boluses of methylprednisolone were associated with a worse prognosis.The beneficial effects seen in critically ill COVID-19 patients treated with corticosteroids might be a class effect. High-dose, rescue boluses seemed harmful.

## 1. Quick Look

### 1.1. Current Knowledge

In critically ill patients with severe forms of COVID-19, administration of corticosteroids is associated with more organ-support-free days and a reduced mortality. Limited anecdotal experience suggested the potential benefit of high-dose boluses of glucocorticoids. Evidence is still lacking as to the impact of the specific type of corticosteroid drug and the effect of boluses.

### 1.2. What This Paper Contributes to Our Knowledge

In a retrospective before/after and propensity-matched case/control study on 81 consecutive, mechanically-ventilated patients with COVID-19-related acute respiratory failure, we could not find any difference between equivalent doses of dexamethasone vs. methylprednisolone in terms of duration of ICU stay, hospital mortality, or incidence of infectious complications. The administration of high-dose corticosteroid boluses was associated with higher hospital mortality, longer mechanical ventilation and ICU and hospital stay, and more infectious complications.

## 2. Introduction

Since SARS-COV-2 initially emerged in China, COVID-19 rapidly spread worldwide. As of today, no specific therapy for SARS-CoV-2 has yet been identified; however, several pre-existing drugs have been suggested for the treatment of infected subjects. Among these, corticosteroids are one of the most debated.

Initially considered contraindicated for concerns of delayed viral clearance and impaired host response [1], corticosteroids have later been suggested as potential key regulators of the hyperinflammatory status responsible for lung damage in the most severe cases [2]. As a matter of fact, imaging of ground glass appearance and histopathologic features of diffuse alveolar damage are consistent with corticosteroid-responsive inflammatory lung disease [3]; moreover, the dysregulated immune response was found to be qualitatively similar to that of multifactorial ARDS [4].

Early case series and retrospective studies [5], and then later large-scale, randomized controlled trials found that, among hospitalized subjects with COVID-19, the use of corticosteroids was associated with an increased number of organ-support-free days [6], and a trend towards a reduction in mortality [7] or treatment failure [8]. The RECOVERY trial finally demonstrated the mortality benefit among subjects receiving oxygen or mechanical ventilation [9]. A subsequent meta-analysis of seven RCTs further confirmed the positive effect of corticosteroids [10].

Nevertheless, evidence is still lacking as to the impact of the specific type of corticosteroid drug [3]. Moreover, anecdotic experience from limited case series suggested the potential beneficial effect of high-dose boluses of corticosteroids in severe cases of COVID-19 [11,12,13].

During the first Italian COVID-19 outbreak (in the spring of 2020), our institution initially decided to treat critically ill subjects with SARS-COV-2-related acute hypoxemic respiratory failure, requiring invasive mechanical ventilation, with a systemic course of dexamethasone, which was in line with the data available at that time [14]; the institutional protocol was then reviewed and methylprednisolone was later suggested as the corticosteroid of choice in those cases, according to the section on ARDS [15] in the ESICM guidelines for the diagnosis and management of critical illness-related corticosteroid insufficiency. On top of the type of corticosteroid selected, the local recommendations allowed clinicians to administer a high-dose, rescue bolus of intravenous methylprednisolone in cases of early refractory hypoxemia, similar to what had been suggested for immunologically mediated lung diseases or SARS [16].

The main outcome of this study was to assess whether the use of dexamethasone vs. methylprednisolone was associated with a different length of ICU stay in critically ill subjects with COVID-19-related acute respiratory failure undergoing mechanical ventilation. Secondary outcomes were the assessment of the effects of rescue, short-term, high-dose boluses of corticosteroids and the analysis of the time course and the factors associated with the response to the boluses of corticosteroids. 

## 3. Methods

### 3.1. Ethics

Ethical approval for this study (Registro Sperimentazioni n. 2020/ST/207) was provided by the Comitato Etico Interaziendale Milano Area 1 by chairperson professor A. M. Di Giulio on 11 November 2020. Written informed consent was obtained according to Italian regulations.

### 3.2. Study Design

A retrospective investigation comparing the outcomes of critically ill COVID-19 subjects receiving two corticosteroid strategies before and after revision of local protocols: a propensity-matched case-control study of subjects who did and did not receive a course of short-term, high-dose boluses of corticosteroids.

### 3.3. Enrolment Criteria

All subjects aged ≥18 years admitted from 1 March to 4 April 2020 to the general ICU of a tertiary care hospital for acute hypoxemic respiratory failure, and with confirmed SARS-CoV-2 infection, were consecutively enrolled. Confirmed infection was defined as a positive reverse transcriptase–polymerase chain reaction from a nasopharingeal swab, associated with symptoms, signs, and radiological findings suggestive of COVID-19 pneumonia. The Berlin criteria for ARDS were used to define and classify the respiratory failure.

### 3.4. Management of Critically Ill Subjects

All subjects were deeply sedated and mechanically ventilated at ICU admission. The clinical management of subjects was standardized according to local and regional suggestion [17]. In particular, the principle of ventilation strategy was early systematic application of the lung protective ventilatory strategy: low tidal volume, medium-high levels of PEEP, and prone position if the PaO_2_/FiO_2_ ratio was <150 mmHg.

### 3.5. Data Collection

Data were retrospectively collected from medical records by trained investigators that were independent from the clinical team. Data on the onset of symptoms, medical history and current medications at time of symptoms onset, clinical and laboratory data at admission, treatment data, and outcome data were collected. Severity scores (SAPS II and SOFA) were calculated at admission, and the clinical frailty scale was used to summarize the overall level of fitness. As for the physiological variables, compliance of the respiratory system was defined as the tidal volume divided by the inspiratory driving pressure. Since alveolar dead space data were not collected, the ventilatory ratio was used as a measure of impaired ventilation [18] where ventilatory ratio is defined as: [minute ventilation (ml/min)  ×  PaCO_2_ (mm Hg)]/(predicted body weight  ×  100  ×  37.5), and higher values indicate more impaired ventilation.

### 3.6. Corticosteroid Regimens

As per institutional protocol, from 1 March 2020 severe subjects requiring ICU admission were given dexamethasone (as an intravenous infusion of 20 mg/day for the first 7 days, followed by 10 mg/day for the following 7 days) on top of the best available supportive care; as of 30 March 2020 the protocol was modified and methylprednisolone (1 mg/kg loading dose, then 1 mg/kg/day continuous infusion for the first 14 days, with the dose halved on the following 14 days) was suggested. On top of that, in case of refractory hypoxemia (defined as a PaO_2_/FiO_2_ ratio < 150 mmHg despite the best supportive care), physicians were allowed to administer a three-day course of 1000 mg/day of methylprednisolone i.v. within the first 10 days of treatment. A positive response to the bolus was defined as any improvement in the PaO_2_/FiO_2_ ratio over the first week after the bolus. All the other aspects in the management of critically ill COVID-19 patients (e.g., indications for intubation, setting of mechanical ventilation, indications for prone positioning, etc.) were unchanged between the two periods.

### 3.7. Outcomes

The main outcome was length of ICU stay; secondary outcomes were duration of mechanical ventilation, length of hospital stay, the number of ventilator-free days during the first 28 days of ICU stay, ICU and hospital mortality, the incidence of ventilator-associated pneumonia, and of bacteremia. 

### 3.8. Statistical Analysis

No missing data for any of the outcomes are present in the dataset; thus, all analyses were complete case analyses. Continuous variables are presented as mean ± standard deviation if normally distributed or medians (25th; 75th quartile) if not; categorical variables are shown as numbers and percentages. Subjects were divided into groups according to the type of corticosteroid received (dexamethasone vs. methylprednisolone), if they did or did not receive a short-term, high-dose bolus of corticosteroids (boluses vs. no boluses), and if they did or did not respond to the boluses (bolus responders vs. non-responders). 

The comparison between dexamethasone and methylprednisolone was considered a before-and-after study. Since the two drugs compared were completely separated in time, and thus potentially subject to time-dependent confounding, a segmented regression was performed to assess if the trends in the length of ICU stay were changing over time independent of the intervention. The study period was divided into consecutive weeks, and the slope and intercept of the regression between the length of ICU stay over time were computed and compared for the two periods in which dexamethasone and methylprednisolone were given. Continuous variables were compared with appropriate parametric or non-parametric tests according to their distribution, and categorical variables were compared with Chi-square tests. The analysis was repeated after adjustment for the baseline imbalance in relevant covariates using appropriate multivariate models.

To account for potential confounders, for the comparison of boluses vs. no boluses, a propensity score was calculated using generalized linear models with a binomial distribution. The probability of a subject receiving a short-term, high-dose course of corticosteroids was estimated as a function of relevant covariates (namely age, SAPS II score, the worst SOFA score excluding the respiratory and liver component during the first 10 days, the worst PaO_2_/FiO_2_ ratio during the first 10 days, the average inspiratory tidal volume, and the worst level of bilirubin during the first 10 days). The results of this logistic propensity model were used to create a nearest-neighbour matched subsample of subjects or for the inverse probability weighting of observations within the final model described above. This allowed the subjects to be weighted based on how likely they were to receive the boluses on the basis of the observed covariates.

In subjects who received the corticosteroid bolus, respiratory mechanics, gas exchange, the ventilator ratio, and SOFA score were assessed and compared at ICU admission, on the day of the bolus administration, and then after 7 and 14 days. A positive response to the bolus was defined as any improvement in the PaO_2_/FiO_2_ ratio over the first week after the bolus. The comparison between responders and non-responders was performed by analysis of variance for repeated measurements, with time as a within-subject factor and the response to the bolus as a fixed, between-subject factor. The model included the interaction effect of time on the response to the bolus. The statistical significance of the within-subject factors was corrected with the Greenhouse–Geisser method. Multiple pairwise, post-hoc comparisons were carried out according to the Tukey method. 

Based on the data from a wide sample of critically ill COVID-19 subjects enrolled in Italy, in which the average length of ICU stay was 12 ± 4 days [19], our retrospective sample of 80 subjects would result in 80% power, at an alpha = 0.05, to detect a 15% reduction in the length of stay between the groups. However, due to the retrospective, low sample size nature of the study, all analyses should be considered exploratory and hypothesis-generating only. The statistical analysis was carried out with STATA version 14.0 (Statacorp, College Station, TX, USA); two-tailed *p*-values < 0.05 were considered for statistical significance.

## 4. Results

A total of 81 subjects were enrolled in the current analysis; Appendix A shows demographic characteristics, comorbidities, treatment received before ICU admission, blood biochemistry, gas exchange, and respiratory physiology at ICU admission. All subjects were intubated and mechanically ventilated at ICU admission. 

A total of 51 subjects (62.9%) received dexamethasone, whereas 29 (35.8%) received methylprednisolone; one subject did not receive any corticosteroid. Table 1 shows the baseline characteristics of subjects in the two groups. As shown, the anthropometric characteristics are comparable, except for a younger age, a higher SOFA score, increased procalcitonin, C-reactive protein and bilirubin, and a higher ventilator ratio at ICU admission in subjects who received dexamethasone.

Appendix A shows the segmented regression for the analysis of the length of ICU stay during the weeks in which dexamethasone and methylprednisolone were used. No differences were found in either the slope or the intercept of the regression curves between the two periods (*p* = 0.786 and *p* = 0.361, respectively). The outcomes of the dexamethasone vs. methylprednisolone comparison are shown in Table 2. Even after correction for the baseline imbalances, no statistically significant differences in hospital mortality or in the duration of mechanical ventilation, the length of ICU stay, or the proportion of subjects who developed infectious complications were found. A higher number of ventilator-associated pneumonia events per subjects and a longer length of hospital stay were found in subjects who received methylprednisolone.

A total of 22 subjects (27.2%) received a rescue course of high-dose corticosteroids during the first 10 days because of refractory hypoxemia. Table 3 shows the baseline characteristics and worst clinical data over the first 10 days of stay in subjects who did vs. those who did not receive the boluses. Subjects who received the boluses had levels of IL-6 that were doubled and a lower oxygenation at baseline, while over the first 10 days of stay they showed a significantly lower oxygenation, a higher driving pressure, and an increased ventilatory ratio.

Table 4 shows the outcomes of subjects who did and did not receive the rescue boluses, both as an unadjusted comparison and after propensity score matching. Subjects who received the rescue boluses had higher hospital mortality, a longer course of mechanical ventilation and of ICU and hospital stay, as well as a significantly higher incidence of infectious complications

Of the 22 subjects who received the boluses, 12 (54.5%) were defined as responders, whereas 10 (45.5%) as non-responders. Figure 1 and Appendix A show the time course of respiratory system compliance, the PaO_2_/FiO_2_ ratio, the SOFA score, and the ventilator ratio at ICU admission, at the time of the bolus, and after 7 and 14 days in subjects classified as responders vs. non-responders. Appendix A shows the baseline characteristics and worst clinical data over the first 10 days of stay in responders vs. non-responders to the rescue boluses; no statistically significant differences were found in any of the available variables. Table 5 shows the outcomes associated with the response to the boluses; subjects who were classified as responders had a significantly lower hospital mortality and a higher number of ventilator-free days.

## 5. Discussion

The main findings of this retrospective, observational study are that: (1) we were unable to observe any difference in subject-oriented outcomes for critically ill subjects admitted to the ICU for COVID-19-related acute respiratory failure who received dexamethasone vs. methylprednisolone, with the exception of a lower length of hospital stay with the use of dexamethasone; (2) subjects who received a course of high-dose, rescue boluses of corticosteroids had a significantly worse outcome; and (3) it may be possible to identify a subgroup of subjects in which the treatment with high-dose boluses of corticosteroids is associated with an improvement in lung mechanics and gas exchange and in which there may be a mortality benefit.

### 5.1. Inflammation and COVID-19

The clinical spectrum of COVID-19 ranges from asymptomatic cases to critical illness with fatal outcomes [19,20,21]. The pathogenesis of COVID-19 appears to be mediated by dysregulated systemic and pulmonary inflammation, along with endothelial injury, hypercoagulability, and thrombosis [22,23]. Platelet–fibrin thrombi formation in small arterial vessels are commonly observed in post-mortem examination of the lungs from subjects with COVID-19 [24]. Emerging data indicate that hypercoagulability in COVID-19 is induced by dysregulated release of neutrophil extracellular traps [25,26,27], and a preclinical investigation found that the administration of dexamethasone was found to reduce the formation of such traps [28]. 

Pulmonary neutrophilia [5] is generally believed to be a key mediator of hypoxemia and ARDS, leading to elaboration of cytokines and chemokines within the pulmonary parenchyma. Corticosteroid therapy aims to support the regulatory function of the activated glucocorticoid receptor α; in subjects with severe COVID-19, glucocorticoid receptor expression in bronchoalveolar lavage myeloid cells is negatively related to lung neutrophilic inflammation and severity of symptoms [29]. The dysregulated immune response observed in COVID-19 is qualitatively similar to that of multifactorial ARDS [4], in whom administration of methylprednisolone was shown to rescue the cellular concentrations and functions of the activated glucocorticoid receptor, leading to downregulation of systemic and pulmonary markers of inflammation, coagulation, and fibroproliferation [15,30].

### 5.2. Corticosteroids and COVID-19

The use of corticosteroids has been debated ever since the first cases of COVID-19. Early expert opinion advised against their use on the basis of pre-existing viral pneumonia literature that showed no obvious benefit and could even potentially cause harm, such as delayed viral clearance [31]. The COVID-19 update of the Surviving Sepsis Campaign guidelines issued a weak recommendation in favor of corticosteroids in mechanically-ventilated subjects with COVID-19, while some panel members preferred not to make a recommendation until further high-quality evidence was presented [32]. On the other side, the Infectious Diseases Society of America guidelines issued a weak recommendation against corticosteroids [1]. Moreover, while the first available case series on COVID-19 from China suggested a potential mortality benefit of corticosteroids [5], previous studies in other viral pneumonias, such as SARS and MERS, found an association with delayed viral clearance, casting concerns that corticosteroids may impair the host response to SARS-CoV-2 [33,34]. However, while viral replication peaks in the second week of illness in SARS-CoV-1 [35], peak shedding in COVID-19 seems earlier [36]. Thus, administration of corticosteroids even early during the hospital stay (however generally after the first week of symptoms) may have no influence on viral replication while at the same time it could reduce the hyper-inflammatory response in COVID-19.

The subsequent, landmark RECOVERY randomized controlled trial demonstrated that daily administration of 6 mg dexamethasone in hospitalized subjects with COVID-19 significantly reduced 28-day mortality and duration of hospital stay, with the greatest mortality reduction observed in those subjects receiving oxygen supplementation or invasive mechanical ventilation [9]. A further, prospective meta-analysis of seven RCTs by the WHO Rapid Evidence Appraisal for COVID-19 Therapies (REACT) further confirmed the benefit of corticosteroid therapy in reducing mortality in critically ill subjects with COVID-19 [10]. The mortality reduction was similar for studies that used dexamethasone or hydrocortisone, suggesting a class effect; however, an optimal dose was not suggested, nor was the clinical threshold for the use of the drug.

While our retrospective study was conducted before the results of the RECOVERY trial [9], our institutional protocol initially suggested a course of dexamethasone for subjects admitted to the ICU with moderate-to-severe ARDS, at a dose similar to a previous positive trial on all-cause ARDS [14]. After an interim revision of the protocol, the type of corticosteroid of choice was modified to methylprednisolone, following the guidance in the section on ARDS [15] in the ESICM guidelines on critical illness-related corticosteroid insufficiency, using a slightly higher equivalent dose based on generally accepted conversion factors.

### 5.3. Case Mix

The current study investigated the effect of different corticosteroid regimens on the clinical outcome of critically ill subjects with COVID-19-related acute hypoxemic respiratory failure.

The subjects enrolled are similar, in terms of demographic characteristics, lung function, and overall outcome, to those previously described in the other ICUs in the Lombardy region [19]; the characteristics are also comparable to subjects enrolled in the RECOVERY trial and, more generally, in the REACT meta-analysis [9,10]. The biochemical profile is also similar to previously published investigations and mainly characterized by increased indices of inflammation and elevated levels of D-dimer [37].

### 5.4. Dexamethasone vs. Methylprednisolone

Subjects who received dexamethasone were younger and had an average lower clinical frailty score; however, they had a higher SOFA score at ICU admission and higher values of inflammation markers, as well as a higher ventilatory ratio. This was likely the result of a more severe form of illness characterized by hyperinflammation and formation of microthrombi in the lung vessels, with a consequent increase in alveolar dead space [22,23,24]. The data likely reflect the different time frame of the epidemic crisis: the first phase, during which the institutional protocol advised the use of dexamethasone, was also the most severe in terms of stress of the healthcare system, when subjects were admitted to the hospitals in more severe condition, and allocation of intensive care treatments was limited by the exceptional, resource-limited, circumstances to subjects with greater chances of therapeutic success [38]. Indeed, even when considering and correcting for such a baseline imbalance, the majority of the patient-centered outcomes explored were not statistically different between the two groups, with the exception of a longer duration of the hospital stay and a higher average number of ventilator-associated pneumonia events for subjects treated with methylprednisolone. This was possibly due to a slightly higher equivalent dose. As the comparison between the two types of corticosteroids includes the effect of time during a dynamic clinical environment, such as the first wave of the COVID-19 surge, we cannot exclude the effect of time-dependent confounding, such as an improvement over time of the outcomes. In an attempt to control for such a bias, a segmented regression was performed to check if the main outcome had a trend over time. Notably, we could not find any significant difference between the periods in which dexamethasone vs. methylprednisolone were given. While no clear differences were identified between the two corticosteroid regimens, if anything, one might infer that the use of dexamethasone might be a more reasonable choice given the similar outcome achieved with a more severe case mix.

### 5.5. Usual Care vs. Rescue Boluses

The two groups were broadly similar in terms of demographic characteristics, biochemistry data at admission, and type of corticosteroid received, the only difference at admission being a higher level of IL-6 in subjects who eventually received a high-dose bolus, a possible expression of a more severe pro-inflammatory state. During the first 10 days of stay, despite a similar respiratory system compliance, subjects who received the rescue bolus not only had significantly worse oxygenation, but also a significantly higher ventilatory ratio. Of note, despite a similar respiratory system compliance, patients who received the bolus had significantly higher airway driving pressure and higher tidal volume, which might have influenced the results. Since we lack data on lung CT scans, it can be argued that patients who received the bolus had a higher extent of fibrotic lesions.

Given the potential selection bias, when comparing the clinical outcomes of subjects who did and did not receive the rescue bolus we built a propensity score based on the available observed covariates. Indeed, even after propensity score matching or weighting the results by the inverse probability of receiving treatment, subjects who received a rescue bolus still had a significantly higher hospital mortality, length of ICU stay and of ventilator assistance, and a higher prevalence of nosocomial infections. 

To the best of our knowledge, this is the largest case series on the use of high-dose boluses of corticosteroids in severe cases of COVID-19. A retrospective observational trial on 92 spontaneously breathing subjects with COVID-19 and cytokine release syndrome assessed the effect of corticosteroid boluses on a composite outcome of endotracheal intubation or death. Subjects received different types of boluses of methylprednisolone (2 mg/kg/day, 250 mg/day, or 500 mg/day for 3 days); of the 92 subjects enrolled, 11 (12.4%) were either intubated or died. Subjects who received the boluses had a non-statistically significant trend towards a reduced incidence of the composite outcome, with no difference among the different doses used [13]. Kolilekas and colleagues reported a small series of six, consecutive, hospitalized COVID-19 subjects with worsening hypoxemia, and indices of hyperinflammatory syndrome, who received a short course of methylprednisolone (125 mg once daily). All subjects developed ARDS between 8 and 13 days after the onset of symptoms. Following the initiation of methylprednisolone, inflammatory markers and oxygenation improved in all subjects and none were intubated [11]. Another case series describes seven subjects with COVID-19-related ARDS who received early treatment with high-dose, short-term boluses of corticosteroids. All subjects received 1000 or 500 mg of methylprednisolone upon intubation, and all were successfully extubated without reintubation and discharged, suggesting that high-dose, short-term corticosteroid therapy early in respiratory failure may provide a good prognosis [12].

However, all these reports were either performed in less severe subjects or did not compare subjects who did with those who did not receive the high-dose boluses of corticosteroids. 

### 5.6. Responders and Non-Responders to the High-Dose, Rescue Bolus of Corticosteroids

As per institutional protocol, clinicians were allowed to administer a short course of high-dose, rescue boluses of corticosteroids in case of early refractory hypoxemia. Given the criteria for the bolus administration, we arbitrarily defined a positive response to the bolus as an increase in oxygenation within one week from administration. Notably, responders were also characterized by improving respiratory system compliance and a reduced ventilatory ratio, as well as a reduced total SOFA score. Moreover, responders had a significantly higher number of ventilator-free days and their hospital mortality nearly halved. Indeed, when we compared bolus responders vs. non-responders, we were not able to find any difference in clinical characteristics or biochemistry neither at admission, nor in respiratory mechanics or gas exchange at the moment of bolus administration. These results seem to suggest that, while the administration of high-dose rescue boluses was associated with worse outcomes in general, a subgroup of subjects exists in which such boluses might be associated with an improved outcome. However, we were not able to find any of the data we gathered which were associated with a positive response to the bolus. It must be noted that the present analysis is limited by the small sample size, which precludes definite conclusions and should be considered only as hypothesis generating.

### 5.7. Limitations

The current study presents several limitations. First, it is a single-center, retrospective, observational study on a relatively limited sample size, and as such it can only be considered as hypothesis generating. Second, the emergency situation which characterized the Lombardy outbreak of COVID-19 deeply limited our ability to collect potentially relevant data which could not be included in the current investigation, such as the inflammatory status at the time of the rescue bolus administration, data on the duration of viral shedding, or CT scan data. Third, we did not record data on adverse effects of the corticosteroid treatment such as the development of hyperglycemia, hypernatremia, ICU-acquired weakness, or delirium. Fourth, the longest available follow up was hospital discharge, and we missed any longer-term outcomes. Fifth, response to steroid administration was defined as any increase in the PaO_2_/FiO_2_ ratio: while there is no clear consensus as how to define a responder in the literature, using different cutoffs might lead to different results. In a post-hoc power analysis, the actual power of the study, based on the effect size estimate from the data, is 0.175; although post-hoc power analyses are not universally recommended, another limitation of our study is that it might have been underpowered.

## 6. Conclusions

In conclusion, in this retrospective investigation on critically ill subjects undergoing invasive mechanical ventilation for COVID-19, we found that the use of dexamethasone, as compared to methylprednisolone, was associated with similar patient-oriented outcomes with the exception of a longer duration of ICU stay. The use of high-dose, rescue boluses of corticosteroids in subjects with refractory hypoxemia was associated with higher ICU and hospital mortality, a longer duration of ventilator assistance, and an increased prevalence of nosocomial infection. A subgroup of subjects in whom the administration of a rescue bolus was associated with an improved outcome was found; however, we were unable to find any factor associated with such a response. In summary, we suggest that a personalized steroid therapy might be the key to make it an effective therapy in COVID-19 ARDS. Further investigations are needed to confirm this finding and to identify subjects who could potentially benefit from a high-dose, rescue bolus of corticosteroids.

## Figures and Tables

**Figure 1 jcm-10-04847-f001:**
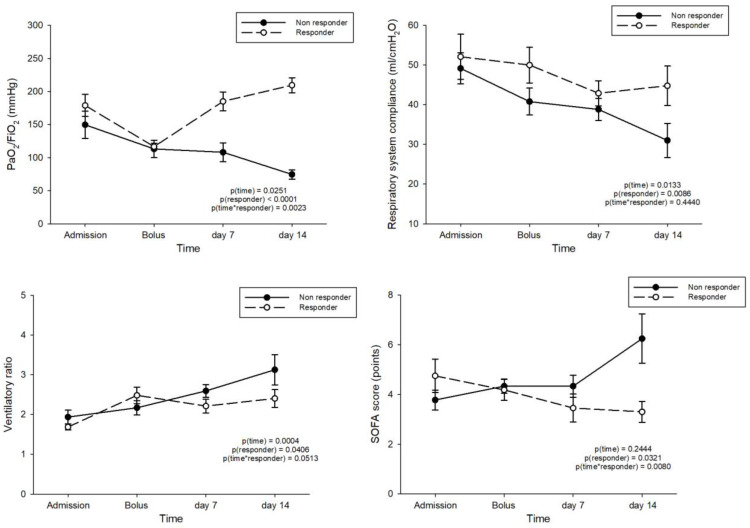
Time course of PaO_2_/FiO_2_ ratio, respiratory system compliance, ventilatory ratio, and the SOFA score at ICU admission, at the time of the bolus, and after 7 and 14 days in patients classified as responders (black dots) vs. non-responders (white dots) to the high-dose bolus of corticosteroids. The comparison between responders and non-responders was performed by analysis of variance for repeated measurements, with time as a within-subject factor and the response to the bolus as a fixed, between-subject factor. The model included the interaction effect of time on the response to the bolus (time*responder). The statistical significance of the within-subject factors was corrected with the Greenhouse–Geisser method.

**Table 1 jcm-10-04847-t001:** Comparison of baseline characteristics of patients who received dexamethasone vs. methylprednisolone.

	Dexamethasone (*N* = 51)	Methylprednisolone (*N* = 29)	*p*
Male sex—*n* (%)	44 (86.3)	24 (82.8)	0.6720
Age—years	58 ± 9	64 ± 9	0.0154
Height (cm)	172 ± 7	173 ± 8	0.7017
Actual body weight (kg)	82 ± 16	80 ± 18	0.6178
Body mass index (kg/m^2^)	27.7 ± 5.3	26.8 ± 5.2	0.4522
Predicted body weight (kg)	67 ± 6	67 ± 6	0.7249
SAPS II score	26 ± 6	28 ± 8	0.2515
SOFA score	5 [3; 7]	3 [2; 4]	0.0057
Frailty score	2 [2; 2]	2 [2; 3]	0.0921
Comorbidities—*n* (%)			
Hypertension	17 (33.3)	13 (44.8)	0.3101
Diabetes	6 (11.8)	4 (13.8)	0.7982
Obesity	14 (27.5)	4 (13.8)	0.1545
COPD	3 (5.9)	1 (3.4)	0.0990
Duration of symptoms before hospitalization (days)	7 [5; 10]	7 [5; 10]	0.6324
CPAP before ICU admission—*n* (%)	47 (92.2)	25 (86.2)	0.3945
Prone position during CPAP—*n* (%)	24 (47.1)	13 (44.8)	0.8471
Duration of CPAP before intubation (days)	2 [1; 4]	3 [1; 6]	0.1232
Prone positioning during ICU stay—*n* (%)	24 (47.0)	10 (34.5)	0.274
Tracheostomy—*n* (%)	17 (33.3)	9 (31.0)	0.8330
Administration of tocilizumab—*n* (%)	12 (23.5)	10 (34.5)	0.2927
Biochemistry at ICU admission			
Procalcitonin (μg/L)	0.53 [0.21; 1.12]	0.26 [0.13; 0.53]	0.0445
C-reactive protein (mg/L)	17.4 [10.9; 24.9]	12.1 [6.3; 17.1]	0.0084
Fibrinogen (mg/100 mL)	680 [610; 770]	660 [514; 747]	0.5111
D-dimer (ng/mL)	2270 [813; 4085]	1225 [712; 5051]	0.6794
Haemoglobin (g/dL)	11.4 ± 1.9	11.2 ± 1.7	0.6765
Platelet count (1/mm^3^)	278,156 ± 11,3686	263,714 ± 88,567	0.5625
White blood cell (1/mm^3^)	9709 ± 4212	8429 ± 3605	0.1786
Creatinine (mg/dL)	0.83 ± 0.37	0.86 ± 0.32	0.8162
Bilirubin (mg/dL)	0.72 [0.49; 1.14]	0.55 [0.39; 0.72]	0.0334
Interleukin-6 (ng/L)	88 [8; 198]	78 [26; 159]	0.9176
Mechanical ventilation and gas exchange at ICU admission			
Respiratory system compliance (ml/cmH_2_O)	50.8 ± 15.6	47.6 ± 14.4	0.3884
Airway driving pressure (cmH_2_O)	10 [8; 12]	11 [9; 13]	0.1734
Airway plateau pressure (cmH_2_O)	24 [22; 26]	24 [23; 28]	0.7076
PEEP (cmH_2_O)	14 [14; 15]	14 [13; 15]	0.1281
Tidal volume (mL)	492 ± 53	491 ± 52	0.9457
Tidal volume (mL/kg abw)	7.4 ± 0.9	7.4 ± 0.7	0.9191
Respiratory rate (1/min)	20 ± 3	19 ± 3	0.5045
FiO_2_ (%)	70.9 ± 13.5	69.6 ± 17.7	0.7247
pH	7.36 ± 0.09	7.37 ± 0.07	0.5994
PaO_2_ (mmHg)	129.6 ± 46.9	135.6 ± 59.8	0.6314
PaCO_2_ (mmHg)	51.2 ± 12.8	46.2 ± 8.5	0.0777
PaO_2_/FiO_2_ (mmHg)	188.6 ± 75.1	201.3 ± 74.5	0.4863
Ventilatory ratio	1.9 [1.5; 2.3]	1.6 [1.3; 2.0]	0.0451

SAPS II: simplified acute physiology score 2nd version; SOFA: sequential organ failure assessment; COPD: chronic obstructive pulmonary disease; CPAP: continuous positive airway pressure; ICU: intensive care unit; PEEP: positive end-expiratory pressure; FiO_2_: fraction of inspired oxygen.

**Table 2 jcm-10-04847-t002:** Comparison of the outcomes of patients who received dexamethasone vs. methylprednisolone.

	Dexamethasone (*N* = 51)	Methylprednisolone (*N* = 29)	*p*	*p* *
Primary outcome:				
Length of ICU stay (days)	13 [7; 21]	11 [6; 22]	0.869	0.272
Secondary outcomes:				
Duration of mechanical ventilation (days)	13 [8; 18]	10 [6; 22]	0.572	0.418
Length of hospital stay (days)	23 [16; 31]	27 [19; 40]	0.101	0.009
Ventilator-free days (days)	0 [0; 19]	15 [0; 20]	0.148	0.931
ICU mortality—*n* (%)	22 (43.1)	11 (37.9)	0.649	0.789
Hospital mortality—*n* (%)	26 (60.0)	11 (37.9)	0.260	0.954
Patients who developed VAP—*n* (%)	25 (49.0)	15 (51.7)	0.816	0.153
Number of VAP per patient	1 [1; 2]	2 [2; 3]	0.006	0.046
Patients who developed bacteremia—*n* (%)	20 (39.2)	10 (34.5)	0.674	0.899
Number of bacteremias per patient	2 [1; 2]	1 [1; 2]	0.770	0.287
High-dose rescue boluses of methylprednisolone—*n* (%)	14 (27.5)	8 (27.6)	0.990	0.908

ICU: intensive care unit; VAP: ventilator-associated pneumonia.* after adjustment for age, SOFA, procalcitonin, C-reactive protein, bilirubin, ventilatory ratio (as per baseline imbalance).

**Table 3 jcm-10-04847-t003:** Comparison of baseline characteristics and worst clinical data over the first 10 days of stay in patients who did vs. those who did not receive a short, rescue course of high-dose boluses of steroids.

	No Boluses (*N* = 59)	Rescue Boluses (*N* = 22)	*p*
Male sex–*n* (%)	50 (84.8)	19 (86.4)	0.8552
Age (years)	60 ± 10	61 ± 8	0.9370
Height (cm)	173 ± 8	172 ± 5	0.5460
Actual body weight (kg)	81 ± 18	83 ± 15	0.6079
Body mass index (kg/m^2^)	27.1 ± 5.3	28.3 ± 5.1	0.3589
Predicted body weight (kg)	67 ± 6	66 ± 4	0.5597
SAPS II score	27.1 ± 6.8	26.3 ± 6.7	0.6316
SOFA score	4 [3; 6]	4 [3; 7]	0.8614
Frailty score	2 [2; 3]	2 [2; 2]	0.6307
Comorbidities–*n* (%)			
Hypertension	22 (39.3)	8 (36.4)	0.8110
Diabetes	9 (16.1)	2 (9.1)	0.4250
Obesity	12 (21.4)	6 (27.3)	0.5811
COPD	2 (3.6)	2 (9.1)	0.3203
Duration of symptoms before hospitalization (days)	7 [5; 10]	7 [5; 10]	0.7977
CPAP before ICU admission–*n* (%)	51 (86.4)	22 (100)	0.0696
Prone position during CPAP–*n* (%)	26 (44.1)	11 (50)	0.6347
Duration of CPAP before intubation (days)	3 [1; 4]	3 [1; 5]	0.3951
Prone positioning during ICU stay–*n* (%)	26 (44.0)	8 (36.4)	0.532
Tracheostomy–*n* (%)	19 (32.2)	7 (31.8)	0.9743
Type of corticosteroid drug–*n* (%)			
Dexamethasone	37 (62.7)	14 (63.6)	0.8284
Methylprednisolone	21 (35.6)	8 (36.4)	
No corticosteroids	1 (1.7)	0	
Aministration of tocilizumab–*n* (%)	14 (23.7)	8 (36.4)	0.2555
Biochemistry at ICU admission			
Procalcitonin (μg/L)	0.42 [0.18; 0.87]	0.36 [0.15; 0.90]	0.6940
C-reactive protein (mg/L)	13.9 [8.5; 20.5]	17.7 [13.2; 23.6]	0.2380
Fibrinogen (mg/100 mL)	664 [587; 747]	685 [619; 786]	0.3774
D-dimer (ng/mL)	1568 [775; 4314]	2419 [712; 4546]	0.4282
Haemoglobin (g/dL)	11.2 ± 1.9	11.7 ± 1.6	0.2940
Platelet count (1/mm^3^)	279,913 ± 112,359	254,818 ± 79,324	0.3405
White blood cell (1/mm^3^)	9442 ± 4228	8746 ± 3389	0.4913
Creatinine (mg/dL)	0.84 ± 0.34	0.88 ± 0.41	0.6075
Bilirubin (mg/dL)	0.7 [0.4; 1.1]	0.6 [0.4; 0.7]	0.1644
Interleukin-6 (ng/L)	58 [20; 142]	198 [91; 264]	0.0258
Respiratory mechanics and gas exchange at ICU admission			
Respiratory system compliance (ml/cmH_2_O)	49.8 ± 15.1	50.4 ± 16.1	0.8713
Airway driving pressure (cmH_2_O)	10 [8; 12]	11 [9; 12]	0.4195
Airway plateau pressure (cmH_2_O)	24 [22; 27]	25 [24; 26]	0.2378
PEEP (cmH_2_O)	14 [12; 15]	14 [14; 14]	0.9575
Tidal volume (mL)	487 ± 55	508 ± 41	0.1143
Tidal volume (mL/kg abw)	7.3 ± 0.8	7.7 ± 0.7	0.0890
Respiratory rate (1/min)	19 ± 3	18 ± 2	0.0582
FiO_2_ (%)	68.9 ± 13.2	73.9 ± 18.3	0.1848
pH	7.36 ± 0.09	7.38 ± 0.08	0.3923
PaO_2_ (mmHg)	137.1 ± 51.2	118.7 ± 49.6	0.1563
PaCO_2_ (mmHg)	50.3 ± 12.7	47.5 ± 8.2	0.3401
PaO_2_/FiO_2_ (mmHg)	204.2 ± 77.4	166.1 ± 58.8	0.0410
Ventilatory ratio	1.9 [1.4; 2.3]	1.8 [1.5; 2.0]	0.6625
Worst respiratory mechanics and gas exchange during the first week			
Respiratory system compliance (ml/cmH_2_O)	47.3 ± 13.0	45.9 ± 13.7	0.6688
Airway driving pressure (cmH_2_O)	11 [8; 12]	13 [11; 16]	0.0065
Airway plateau pressure (cmH_2_O)	24 [22; 27]	27 [22; 29]	0.1921
PEEP (cmH_2_O)	14 [12; 15]	14 [12; 14]	0.8115
Tidal volume (mL)	485 ± 54	546 ± 81	0.0003
Respiratory rate (1/min)	20 ± 3	20 ± 2	0.5984
FiO_2_ (%)	69.1 ± 12.9	70.7 ± 14.6	0.6516
pH	7.35 ± 0.08	7.40 ± 0.07	0.0239
PaO_2_ (mmHg)	126.3 ± 40.2	80.7 ± 20.9	<0.0001
PaCO_2_ (mmHg)	50.8 ± 12.2	52.0 ± 10.0	0.6684
PaO_2_/FiO_2_ (mmHg)	187.5 ± 60.2	116.3 ± 28.2	<0.0001
Ventilatory ratio	1.9 [1.5; 2.3]	2.1 [1.8; 3.0]	0.0445
Worst SOFA score	4 [3; 6]	4 [4; 6]	0.1162

SAPS II: simplified acute physiology score 2nd version; SOFA: sequential organ failure assessment; COPD: chronic obstructive pulmonary disease; CPAP: continuous positive airway pressure; ICU: intensive care unit; PEEP; positive end-expiratory pressure; FiO_2_: fraction of inspired oxygen.

**Table 4 jcm-10-04847-t004:** Comparison of the outcomes of patients who did and did not receive a rescue course of high-dose corticosteroids.

	No Boluses	Rescue Boluses	Regression Output	*p*
(a) Unadjusted comparison	*N* = 59	*N* = 22		
Primary outcome				
Length of ICU stay (days)	10.5 [6; 17]	18 [13; 23]	b = 6.69 ± 3.35	0.050
Secondary outcomes				
Duration of mechanical ventilation (days)	9.5 [6; 16]	18 [13; 23]	b = 7.66 ± 2.86	0.090
Length of hospital stay (days)	25 [18; 35]	21 [16; 37]	b = 0.88 ± 3.92	0.822
Ventilator-free days (days)	15 [0; 22]	0 [0; 0]	b = −9.93 ± 2.34	<0.001
Hospital mortality–*n* (%)	20 (33.9)	17 (77.3)	OR = 6.63 ± 3.83	<0.001
Patients who developed VAP–*n* (%)	24 (40.7)	16 (72.7)	OR = 3.88 ± 2.13	0.013
Number of VAP per patient	2 [1; 2]	2 [1; 2]	b = 0.25 ± 0.37	0.505
Patients who developed bacteremia–*n* (%)	16 (27.1)	15 (68.2)	OR = 5.75 ± 3.12	<0.001
Number of bacteremias per patient	2 [1; 2]	1 [1; 2]	b = −0.21 ± 0.27	0.434
(b) Propensity-matched sample	*n* = 22	*N* = 22		
Primary outcome				
Length of ICU stay (days)	11 [6; 24]	18 [13; 23]	b = 14.61 ± 7.60	0.059
Secondary outcomes				
Duration of mechanical ventilation (days)	9 [6; 18]	18 [13; 23]	b = 15.85 ± 7.57	0.040
Length of hospital stay (days)	27 [21; 35]	21 [16; 37]	b = 9.29 ± 7.26	0.205
Ventilator-free days (days)	8 [0; 21]	0 [0; 0]	b = −9.30 ± 2.08	<0.001
Hospital mortality–*n* (%)	9 (40.9)	17 (77.3)	OR = 9.84 ± 7.67	0.003
Patients who developed VAP–*n* (%)	11 (50.0)	16 (72.7)	OR = 9.68 ± 6.90	0.001
Number of VAP per patient	1 [1; 2]	2 [1; 2]	b = 1.80 ± 1.08	0.105
Patients who developed bacteremia–*n* (%)	5 (22.7)	15 (68.2)	OR = 16.62 ± 11.7	<0.001
Number of bacteremias per patient	2 [2; 3]	1 [1; 2]	b = 0.19 ± 0.52	0.710

ICU: intensive care unit; VAP: ventilator-associated pneumonia.

**Table 5 jcm-10-04847-t005:** Comparison of the outcomes of patients who were classified as responders vs. non-responders to the rescue course of high-dose corticosteroids.

	Bolus Non-Responders (*n* = 10–45.5%)	Bolus Responders (*n* = 12–54.5%)	*p*
Pimary outcome			
Length of ICU stay (days)	21 [11; 23]	18 [16; 26]	0.8868
Secondary outcomes			
Duration of mechanical ventilation (days)	21 [12; 23]	18 [14; 23]	0.9150
Length of hospital stay (days)	21 [15; 23]	33 [17; 46]	0.2703
Ventilator-free days (days)	0 [0; 0]	0 [0; 9]	0.0324
Hospital mortality–*n* (%)	10 (100)	7 (58.3)	0.0274
Patients who developed VAP–*n* (%)	7 (70.0)	9 (75.0)	0.8825
Number of VAP per patient	2 [1; 2]	2 [1; 2]	0.7665
Patients who developed bacteremia–*n* (%)	7 (70.0)	8 (66.7)	0.5773
Number of bacteremias per patient	1 [1; 2]	2 [1; 2]	0.5085

ICU: intensive care unit; VAP: ventilator-associated pneumonia.

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
