# Peer review of "Effect of Different Corticosteroid Regimens on the Outcome of Severe COVID-19-Related Acute Respiratory Failure. A Retrospective Analysis"

_jcm, 2021, doi:10.3390/jcm10214847_

Round 1
Reviewer 1 Report
The authors report here the results of a before-and-after study comparing Covid 19 patients receiving either dexamethasone or methylprednisolone. To my opinion the data reported are original, the methodology seems robust.
My comments are as follows:
- Usual care vs rescue bolus: the authors observed a higher mortality in the patients who received rescue bolus, even after building a propensity score based on available covariables. It may be argued that there were no CT scan data in the two groups. Perhaps the group who received bolus had more fibrotic lesions. Also it seems that the group who received bolus was more severe in terms of respiratory mechanics before the administration of bolus: compliance was similar but driving pressure was higher, also ventilatory ratio, tidal volume was clearly higher and likely over 8 ml/kg abw. This needs to be discussed.
- Responders vs non-responders to rescue bolus: the limitations of this analysis must be emphasized, mainly the little sample: 12 vs 10 patients. Moreover, the authors need to define precisely the response in PaO2/FIO2. It is usually defined as an improvement of PaO2/FIO2 of more than 20%. Defining the response as an absolute improvement of PaO2/FIO2 seems me questionable. Please clarify.
Author Response
The authors report here the results of a before-and-after study comparing Covid 19 patients receiving either dexamethasone or methylprednisolone. To my opinion the data reported are original, the methodology seems robust.
R: we thank this reviewer for his comments
My comments are as follows:
- Usual care vs rescue bolus: the authors observed a higher mortality in the patients who received rescue bolus, even after building a propensity score based on available covariables. It may be argued that there were no CT scan data in the two groups. Perhaps the group who received bolus had more fibrotic lesions. Also it seems that the group who received bolus was more severe in terms of respiratory mechanics before the administration of bolus: compliance was similar but driving pressure was higher, also ventilatory ratio, tidal volume was clearly higher and likely over 8 ml/kg abw. This needs to be discussed.
R: these interesting issue were added into the discussion section (line 390-394 page 13 “Of note, despite a similar respiratory system compliance, patients who received the bolus had a significantly higher airway driving pressure and a higher tidal volume, and this might have influenced the results: since we lack data on lung CT scan, it can be argued that patients who received the bolus had a higher extent of fibrotic lesions”) and the lack of CT scan data was added in the limitations
- Responders vs non-responders to rescue bolus: the limitations of this analysis must be emphasized, mainly the little sample: 12 vs 10 patients.
R: this was added as a limitation (page 14 line 437-438 “It must be noted that the present analysis is limited by the small sample size, which precludes definite conclusions and should be considered only as hypothesis-generating.”)
Moreover, the authors need to define precisely the response in PaO2/FIO2. It is usually defined as an improvement of PaO2/FIO2 of more than 20%. Defining the response as an absolute improvement of PaO2/FIO2 seems me questionable. Please clarify.
R: The response to steroid administration was defined as any increase in the PaO2/FiO2 ratio. This was clarified in the methods (page 4 line 150-151 “A positive response to the bolus was defined as any improvement in the PaO2/FiO2 ratio over the first week after the bolus “) and added as a limitation (page 14 line 449-455: “Fifth, response to steroid administration was defined as any increase in the PaO2/FiO2 ratio: while there is no clear consensus as how to define a responder in the literature, using different cutoffs might lead to different results.”). Indeed, it must be acknowledged that there is no clear consensus as how to define “responders”. This has variably been done in the literature: from absolute increases (Langer, Chest 1988 DOI:10.1378/chest.94.1.103 defined a PaO2 increase of at least 10 mmHg; Pelosi, MAS 2001 used at least 20 mmHg increase in the P/F ratio) to relative increases (Weiss, BJA 2021 DOI:10.1016/j.bja.2020.09.042 used at least 20% increase, while Papazian, Anesthesiology 2002 used a 33% cutoff).
Reviewer 2 Report
In this interesting study by Umbrello et al, two steroid regimens for COVID-19 critically ill patients were compared retrospectively. The topic is interesting, but the study design and the small sample size limit its clinical meaning. The post hoc power analysis is mandatory to understand to real power of the study since the risk for it to be underpowered is high. The main message of the study is that in personalizing steroid therapy may be the key to make it an effective therapy in COVID-19 ARDS. Specific comments: Methods: - Page 4, lines 13-14. The protocol for steroid treatment was changed; was there any other change in the treatment protocol for these patients, e.g. indication for prone positioning or for intubation? - Page 5, 19: Please calculate the actual power of the study (post hoc power analysis) by using the data form your study. It seems that the power provided (80% in the detection of 15% difference) may be overrated. - Please explain better the definition of refractory hypoxemia for extra bolus administration. Was prone positioning considered as a therapeutic option? - which was the protocol for mechanical ventilation? - Did the patients fulfill the Berlin criteria for ARDS? Results - Page 7, line 24: please define better responders and non-responders. - What about prone positioning in the two groups? Which was the protocol for prone positioning? Was it performed? - Did the authors find any difference in the glycemia levels between the two groups? What about the bolus group? Discussion - Please provide some clinical message from the data of your studyAuthor Response
In this interesting study by Umbrello et al, two steroid regimens for COVID-19 critically ill patients were compared retrospectively. The topic is interesting, but the study design and the small sample size limit its clinical meaning. The post hoc power analysis is mandatory to understand to real power of the study since the risk for it to be underpowered is high. The main message of the study is that in personalizing steroid therapy may be the key to make it an effective therapy in COVID-19 ARDS.
Specific comments:
Methods: - Page 4, lines 13-14. The protocol for steroid treatment was changed; was there any other change in the treatment protocol for these patients, e.g. indication for prone positioning or for intubation?
R: all the other aspects of the treatment of these patients, but the corticosteroid regimen, were unchanged. This was added in the methods section (page 4 line 151-154 “All the other aspect in the management of critically ill, COVID-19 patients (eg. indications for intubation, setting of mechanical ventilation, indications for prone positioning, etc.) were unchanged between the two periods.”
- Page 5, 19: Please calculate the actual power of the study (post hoc power analysis) by using the data form your study. It seems that the power provided (80% in the detection of 15% difference) may be overrated.
R: We performed the post-hoc power analysis, and presented the results and its implications in the discussion section (page 14, line 452-455 “In a post-hoc power analysis, the actual power of the study based on the effect size estimate from the data is 0.175; although post-hoc power analyses are not universally recommended, another limitation of our study is that it might have been underpowered.”
- Please explain better the definition of refractory hypoxemia for extra bolus administration.
R: As stated in the methods section (page 4 line 147-148), refractory hypoxemia was defined as “ a PaO2/FiO2 ratio <150 mmHg despite the best supportive care”. This definition was taken from Collins et al, Respir care 2011
Was prone positioning considered as a therapeutic option?
R: According to the available recommendations at the time the study was performed (see Foti et al. Minerva Anestesiologica 2020), prone positioning was recommended for all the patients with a PaO2/FiO2 ratio <150 mmHg. This was stated in the methods section
- which was the protocol for mechanical ventilation?
R: Again, we followed standardized, regional recommendations (Foti et al. Minerva Anestesiologica 2020). We added these details in the methods section: page 3 line 125-127 “In particular, the principle of ventilation strategy was early systematic application of the lung protective ventilatory strategy: low tidal volume, medium-high levels of PEEP, prone position if the PaO2/FiO2 ratio was <150 mmHg”
- Did the patients fulfill the Berlin criteria for ARDS?
R: All the patients fulfilled the Berlin Criteria, and this was stated in the methods section: page 3 lines 120-121 “The Berlin criteria for ARDS were used to define and classify the respiratory failure.”
Results
- Page 7, line 24: please define better responders and non-responders.
R: The response to steroid administration was defined as any increase in the PaO2/FiO2 ratio. This was clarified in the methods (page 4 line 150-152 “A positive response to the bolus was defined as any improvement in the PaO2/FiO2 ratio over the first week after the bolus “) and added as a limitation (page 14 line 450-452: “Fifth, response to steroid administration was defined as any increase in the PaO2/FiO2 ratio: while there is no clear consensus as how to define a responder in the literature, using different cutoffs might lead to different results.”).
- What about prone positioning in the two groups? Which was the protocol for prone positioning? Was it performed?
R: Prone positioning was indicated when the PaO2/FiO2 ratio was <150 mmHg; it was performed in 18h-cycles, and was prosecuted until the PaO2/FiO2 ratio increased >150 mmHg. More detailed data about prone positioning in the different groups was provided in the tables 1 and 3
- Did the authors find any difference in the glycemia levels between the two groups? What about the bolus group?
R: We thank this reviewer for his insightful comment. Unfortunately, blood glucose levels, although tightly monitored, were not recorded for the purposes of the current investigation. This was added as a limitation
Discussion
- Please provide some clinical message from the data of your study
R: a sentence was added in the conclusion, which states “In summary, we suggest that a personalized steroid therapy might be the key to make it an effective therapy in COVID-19 ARDS”
Round 2
Reviewer 1 Report
The authors have responded to all my comments on the previous version of the manuscript, and the modifications provided seem me adequate.